# Associations Between Maximal Passive Knee Extension and Sagittal Plane Kinematic Patterns in Children with Spastic Cerebral Palsy: A Longitudinal Study

**DOI:** 10.3390/jcm14238567

**Published:** 2025-12-03

**Authors:** Inti Vanmechelen, Edwin Råsberg, Eva Broström, Cecilia Lidbeck

**Affiliations:** Department of Women’s and Children’s Health, Karolinska Institutet, 171 76 Stockholm, Sweden; edwin.rasberg@ki.se (E.R.); eva.brostrom@ki.se (E.B.); cecilia.lidbeck@ki.se (C.L.)

**Keywords:** cerebral palsy, three-dimensional gait analysis, statistical parametric mapping, passive knee range of motion

## Abstract

**Highlights:**

**What are the main findings?**
A relatively small limitation of passive knee extension was associated with knee flexion during gait that strengthened with age and was more prominent in bilateral CPKnee flexion during gait remained stable over time despite the increasing number of children with knee contractures

**What is the implication of the main finding?**
Knee flexion during walking is important to consider for early-stage and effective treatment planning, aiming to prevent possible development of knee contractures in children with bilateral CP

**Abstract:**

**Background/Objectives:** There is limited information on the interplay between passive joint motion and joint kinematics from three-dimensional gait analysis (3DGA) and its longitudinal evolution in cerebral palsy (CP). We aimed to associate clinical measurements and gait kinematics over time using a longitudinal study design. **Methods**: Ambulatory individuals with spastic CP, aged 4–18, who performed a minimum of two 3DGA at the Karolinska University Hospital between 2008 and 2025 were recruited. Primary outcomes were sagittal plane kinematics and maximum passive knee extension (pKE). Canonical correlation (R) with statistical parametric mapping was used to associate passive maximum knee extension with sagittal hip, knee, and ankle angles at two timepoints. **Results**: the 3DGA data of 31 children (age 4–17 years; mean age 10.4 +/− 2.9) with 22 bilateral (bCP, GMFCS I = 6; II = 13; III = 3) and 9 unilateral CP (uCP, GMFCS I = 8; II = 1) was included. For the whole and bCP groups, respectively, knee flexion/extension and pKE were correlated throughout stance (*p* < 0.001), with R between −0.47 and −0.57/−0.49 and −0.59 at T1 and between −0.46 and −0.72/−0.50 and −0.76 at T2. Hip flexion/extension and knee pKE were correlated from 17 to 62%/46–52% of the gait cycle (*p* < 0.001/*p* = 0.045) for the whole and bCP groups, respectively, with R between −0.41 and −0.57/−0.38 and −0.41 at T1 and from 15 to 64%/17 to 61% with R between −0.50 and −0.57/−0.42 and −0.57 at T2. **Conclusions**: Reported associations between structural knee properties and knee position during gait demonstrated progression over time, implying that a restricted range of motion may be driven by functional constraints. Combining knee contractures and their longitudinal development with 3DGA is a powerful approach for pre-intervention planning.

## 1. Introduction

In children with cerebral palsy (CP)—a disorder of the development of movement and posture—abnormal gross and fine motor functioning have an influence on mobility, functional activities, and gait [1]. The motor disorders of CP are often accompanied by musculoskeletal constraints such as reduced passive range of motion, spasticity, and weakness [1]. Within CP, spasticity is the most frequently represented neurological feature, and the spastic subtype accounts for approximately 77% of the population [2]. Spastic CP is further classified as either bilateral (bCP) or unilateral (uCP) [3]. Even though gross motor functioning varies, more than fifty percent of children with CP walk without assistive devices [4]. Reduced walking ability and community participation, as well as pain, have a significant impact on quality of life in individuals with CP [5,6].

Due to the complex nature and individualized symptoms in CP, interprofessional management is recommended when planning interventions aimed at enhancing walking ability. Three-dimensional gait analysis (3DGA) is often used to evaluate the effect of interventions on gait quality in children with CP. 3DGA yields objective information on kinematics, i.e., pelvis, hip, knee, and ankle angles during a gait cycle in the stance phase and the swing phase. The importance of 3DGA, in combination with information from clinical examination and anamnesis, has long been emphasized [7,8,9] and was confirmed by a study from Schwartz and colleagues. They found that instrumented gait analysis using 3DGA tends to recommend fewer surgical procedures than initial clinical plans, especially when interprofessional interventions are considered [10].

Various interventions have shown positive effects on passive joint range of motion and/or kinematic patterns. Ounpuu and colleagues assessed the long-term (11-year) effects of multilevel surgery in 51 adults with CP and found that knee flexion at initial contact and peak ankle dorsiflexion in stance were maintained, but range of motion improvements were not maintained beyond one year post-surgery [11]. Verreydt et al. found no range of motion improvements one year after a selective dorsal rhizotomy in 15 children with CP, but the total gait profile score and the gait profile score at the level of the ankle and knee did improve post-intervention. The reported effects of botulinum toxin A on clinical and gait parameters are variable, ranging from no effect [12] to significant improvements in knee position in midstance [13] and short-term improvements in the kinematics patterns of the ankle and the knee [14].

While the importance of 3DGA for clinical decision making and assessment of interventions has been well established, previously reported associations between clinical assessments and parameters extracted from 3DGA were low. Desloovere et al. found that correlations between passive hip and knee range of motion and maximal knee and ankle angle during different phases of the stance phase ranged from −0.14 to 0.50 [15]. Similarly, McMulkin and colleagues reported associations between passive knee extension and knee extension at initial contact, and minimal knee flexion of −0.24 and −0.45, respectively [16].

The varying above-mentioned outcomes imply that the interplay between clinical assessments and joint kinematics is complex and might require the consideration of longitudinal data, whereas previous associations were all assessed at cross-sectional timepoints. Specifically for range of motion and muscle lengths, it is currently unknown whether reduced passive range of motion contributes to a flexed knee gait pattern in stance or whether children with reduced knee ranges during walking may tend not to use their available passive range of motion [17,18]. Understanding the development of both passive range of motion and its influence on joint angles during gait could improve insights into how intervention strategies, such as soft-tissue surgeries or botulinum toxin A type-A injections, may or may not improve joint ranges during walking.

With respect to longitudinal data from 3DGA, O’ Sullivan and colleagues found no differences in mean knee angle at mid-stance between baseline and follow-up (minimum two years apart) in a cohort of 27 children with bilateral CP [19]. On the other hand, Daly and colleagues found significant differences between baseline and follow-up for the sagittal knee angle in midstance in 180 children with bilateral CP, with reduced knee extension in follow-up [20]. One must note that the cohort in Daly’s study was much larger than in O’ Sullivan’s study, which may have influenced the results. These varying findings support the hypothesis that the relationship between clinical measures and kinematic joint angles is complex and requires the use of advanced methodologies.

Statistical Parametric Mapping (SPM) is such a method, accommodating the comparison of time-dependent signals, such as kinematic/kinetic gait patterns, where, e.g., joint angles are registered over a period of time [21]. Traditional statistical comparisons (e.g., *t*-tests or ANOVA) require selecting specific timepoints at which outcome measures are compared between conditions or groups. This approach causes a loss of information—there may not be any differences in the pre-chosen variables and significant differences at other timepoints may be missed—and may contain bias, since the parameters of interest to the researchers will be chosen most often. Additionally, SPM also allows associations between a time-dependent signal—e.g., knee flexion/extension angle throughout the gait cycle—and discrete values—e.g., range of motion or spasticity. Recently, Papageorgiou and colleagues used SPM in a cohort of children with CP to explore associations between outcomes from the physical examination and gait deviations at the hip, knee, and ankle [22]. In children with bilateral CP (bCP), lack of passive knee extension was associated with decreased knee extension and decreased ankle dorsiflexion throughout stance, whereas this was not the case in children with unilateral CP (uCP) [22]. The association at the level of the ankle indicates that reduced passive knee extension could be a contributing factor to deviating patterns that go beyond the level of the knee.

Papageorgiou et al. included a large sample size with a wide age span, ranging from three to eighteen years of age [22]. This broad age range complicates the interpretation of the associations, because the effect of normal growth cannot be modeled in a cross-sectional study design. Daly and colleagues reported an age-related progression of physical measures around the hip and knee as well as gait kinematics, including reduced hip and knee extension and increased ankle dorsiflexion in stance [16]. These findings emphasize the importance of the natural progression of clinical and kinematic variables in children with CP [20]. Additionally, knee contractures were shown to develop differently with age and may thus also have varying effects on gait patterns [23].

Considering the importance of growth and development and the lack of information about the longitudinal relationships between passive joint range of motion and kinematic angles at different joint levels during walking, the current study aimed to associate clinical assessments and gait kinematics using longitudinal assessments. The primary objective was to explore the associations between maximal passive knee extension and the sagittal plane joint kinematics of the hip, knee, and ankle throughout the gait cycle at two timepoints during childhood for each individual with CP. To explore the natural progression of gait patterns with time, a secondary objective was to compare the sagittal plane joint kinematics between timepoints.

## 2. Materials and Methods

### 2.1. Participants

The sagittal plane gait data of children with spastic CP who visited the Motion analysis Laboratory at Karolinska University Hospital for a clinical gait analysis between 2008 and 2025 was included in this retrospective study. Inclusion criteria were (i) a confirmed CP diagnosis, (ii) unsupported walking ability, i.e., Gross Motor Function Classification System (GMFCS) level I, II, or III, with GMFCS describing motor functioning, focusing on self-mobility and transfer, at five levels, ranging from walking without limitations at Level I to being transported in wheelchairs at Level V [24], (iii) age between four and eighteen years, (iv) a minimum of two 3DGA sessions with at least one year in between, and (v) a minimum of two good barefoot kinematic trials for each session. Data was excluded if the child used a walker and/or assistive device. Data from both legs was included for children with bCP, and data from the most affected leg in children with uCP. For participants with more than two visits, the first and last visits with availability of barefoot independent walking data were consistently selected. All testing was performed by experienced physiotherapists. This study was conducted in accordance with the Declaration of Helsinki and approved by the Regional Ethics Board in Stockholm, Sweden (No. 2021-04435). Written informed consent was obtained from the parents and/or the participating children.

### 2.2. Clinical Examination

All participants underwent a clinical examination, including measurements of maximum passive range of motion of the hip, knee, and ankle with a goniometer in a supine position by a specially trained, experienced physiotherapist, following a standardized protocol [25]. Reduced maximum passive knee extension in the knee was defined as passive range of motion below a neutral position of the knee joint.

### 2.3. Three-Dimensional Gait Data

During the three-dimensional gait analysis, the child walked barefoot on a 10 m walkway at a self-selected walking speed until three trials with good force plate hits were obtained for clinical purposes. However, force plate information was not used in the current study. Reflective markers (diameter: 10 mm) were attached to the skin according to the Plug-In Gait lower limb model, using 16 reflective markers placed on anatomical landmarks [26,27,28]. Marker trajectories were sampled at 100 Hz with an 8 to 12 Vicon camera system (Vicon-UK, Oxford, UK). A built-in Woltring filter using mean square error and a 10 mm^2^ smoothing factor was used to filter the trajectories. Gait cycles were identified from initial contact and foot-off events, and hip, knee, and ankle kinematics were calculated in the Vicon Nexus software (Version 2.12.1), using a mean square error Woltring filter with a smoothing factor of 10 mm^2^. The kinematic waveforms were time-normalized, yielding a total of 101 data points for each curve, and the quality of the gait trials was visually checked in a custom-made Python software (Python Software Foundation, version 2.7; https://www.python.org/). If two or three gait cycles were available for each individual, gait trials were averaged such that one averaged gait cycle per individual was included in the analyses.

To strive for valid and reliable data and limit the risk of uncontrolled variability, routines for gait analysis recordings, processing of data, and clinical examinations are standardized following the standard of the European Society of Motion Analysis. Equipment was updated with new and more cameras (Vicon-UK, Oxford, UK). during the period of data collection.

### 2.4. Statistical Analysis

For the descriptive data, values were presented as numbers, specifically the mean values. Parametric tests in SPSS (version 30.0.0.0; Armonk, NY, USA) were used for within-group analysis over time. Within the SPM technique, regression analysis was used to assess the correlations between maximal passive knee extension (discrete parameter) and joint angular waveforms (continuous parameters) (SPM1d version 0.4, available for download at http://www.spm1d.org/) in Python. The random field theory applied in SPM produces a critical threshold, which is the value that needs to be exceeded for the waveforms to obtain significant differences at the 0.05 level [21]. Subsequently, supra-threshold clusters are identified, i.e., the time periods in the waveform for which the data exceeded the critical threshold. In this case, the information concerning the *p*-value, the extent (percentage of the gait cycle), and the location (first and last points of the cluster) was extracted and reported. The alpha-level for all correlation analyses was set to 0.025, considering correlations were performed for both the whole group and the bCP group. Additionally, the strength of the correlation (R) throughout the gait cycle was added to the SPM plot to allow interpretation of the strength of the correlation. The following interpretation was used for the size of the correlation: 0.00–0.30, negligible correlation, 0.30–0.50, low correlation, 0.50–0.70, moderate correlation, 0.70–0.90, high correlation, and 0.90–1.00, very high correlation [29]. R^2^—the coefficient of determination—was additionally used to interpret the clinical strength of the found associations, as this value indicates the proportion of variance that is shared by two variables [30]. An example of the method is depicted in Figure 1 for the association between the angular waveforms of knee flexion/extension angle and maximal passive knee extension. The alpha-level was adjusted for the degrees of freedom of each joint (three for knee, hip, and ankle). As the data was normally distributed, parametric SPM results for all outcome measures were reported.

GenAI has been used in this paper to assist in data analysis. More specifically, Microsoft Co-Pilot (Microsoft Corporation; version 1.25111.85.0) was used to assist in writing the code to normalize kinematic data and generate the figures included in the manuscript. All outputs from GenAI were manually verified to ensure correct outputs. No GenAI was used in the writing of the manuscript.

## 3. Results

Participant characteristics for the total group, the bCP group, and the uCP group can be found in Table 1. Figure 2 schematically depicts all participants and the assessments performed.

### 3.1. Participants

Retrospective kinematic gait data of 31 children with spastic CP (22 bCP, 9uCP; 12 females) was included in the analysis. Participants’ ages ranged from four to eleven years (mean age 8.2 +/− 2.0 years) at assessment point 1 (T1) and from five to seventeen (mean age 12.4 +/− 2.8 years) at assessment point 2 (T2). The time between both assessments ranged from one to seven years (median 3.9 years). Motor functioning of the participants in the bCP group ranged from GMFCS I to III (I = 6; II = 13; III = 3), whereas the uCP group was at GMFCS I (8) and II (1). For the bCP group, data from both legs were included (yielding 44 legs), and for the uCP group, 5 left and 4 right legs were included.

### 3.2. Maximal Passive Knee Extension

For the bCP group, 9 out of 44 legs showed reduced passive knee extension at T1 and 21 out of 44 at T2. For the uCP group, only 1 affected leg showed reduced passive knee extension at T2. The median passive knee extension for the whole, bCP, and uCP groups at both T1 and T2 was 0. The mean passive knee extension for the whole group was −0.40 at T1 and −4.5 degrees at T2. For the bCP group, this was −0.56 and −5.66, and for the uCP group, this was 0 and 0.55. There were no significant differences between passive knee extension at T1 and T2.

### 3.3. Kinematics of the Knee During Gait

The median maximal knee extension in stance in the whole CP group was 9.12 degrees at T1 and 10.2 degrees at T2. For the bCP group, this was 10.78 degrees at T1 and 12 degrees at T2. In accordance with gait analysis standards, full knee extension is 0 degrees, positive values reflect knee flexion, and negative values reflect knee hyperextension. Neither group had significantly different maximal knee extension values over time.

### 3.4. Associations

For associations between passive knee extension and hip, knee, and ankle kinematics, the whole group (bCP + uCP) and the bCP group were analyzed separately. Since only one participant in the uCP group presented with a passive knee extension deficit (at T2), no associations were performed for the uCP group separately.

### 3.5. Associations Between Passive Knee Extension and Hip Kinematics

For the whole group (n = 53 legs) and the bCP group (n = 44 legs), there were no significant associations between hip flexion/extension during gait and passive knee extension at T1 (Figure 3A,C). At T2, there was a significant association between hip flexion/extension and passive knee extension from 15 to 64% (*p* < 0.001) of the gait cycle, with R between −0.41 and −0.57 for the whole group (Figure 3B). For the bCP group, there was a significant association between hip flexion/extension and passive knee extension from 17 to 61% (*p* < 0.001) of the gait cycle, with R between −0.42 and −0.57 at T2 (Figure 3D).

### 3.6. Associations Between Passive Knee Extension and Knee Kinematics

For the whole group (n = 53 legs), there was a significant association at T1 between knee flexion/extension during gait and passive knee extension from 3 to 68% of the gait cycle (*p* < 0.001), with R between −0.47 and −0.56 (Figure 4A). At T2, there was a significant association between knee flexion/extension and passive knee extension from 0 to 63% (*p* < 0.001) and 85 to 99% (*p* < 0.028) of the gait cycle, with R between −0.46 and −0.66 (Figure 4B). For the bCP group (44 legs), there was an association between knee flexion/extension and passive knee extension at T1 from 4 to 68% of the gait cycle (*p* < 0.001) with R between −0.49 and −0.59 (Figure 4C). At T2, there was a significant association between knee flexion/extension and passive knee extension from 0 to 65% (*p* < 0.001) and 86 to 99% (*p* < 0.033) of the gait cycle, with R between −0.50 and −0.76 (Figure 4D).

Considering the greater association between passive knee extension and the sagittal knee angles at T2, we additionally looked at the gait patterns of the individuals with bCP who did not have a passive knee extension deficit at T1, but developed a knee contracture towards the second assessment, T2. A total of 11 individuals with bCP developed a knee contracture between the first and second assessment, and none of these individuals obtained a full knee extension in stance phase at T1 (Figure 5).

### 3.7. Associations Between Passive Knee Extension and Ankle Kinematics

For both the whole group and the bCP group, there were no associations between ankle plantar/dorsiflexion during gait and passive knee extension at any timepoint (Appendix A).

### 3.8. Kinematic Patterns over Time

Hip flexion/extension during gait differed significantly over time. Hip flexion increased between T1 and T2 for the whole CP group (16–58%; *p* = 0.001), and for the bCP group (15–58%; *p* = 0.001). The knee flexion/extension angle during gait did not differ significantly between T1 and T2 in either group. For the ankle dorsiflexion/plantarflexion angle, there was a significant difference over time. Dorsiflexion in swing increased for the whole CP group (64–98%; *p* = 0.001), and for the bCP group (66–84%; *p* = 0.016 and 89–97%; *p* = 0.040) (Figure 6A–I).

### 3.9. Kinematic Patterns with Respect to Intervention

We additionally added the effect of intervention for the bCP group to discuss this factor in a descriptive way, since the group numbers were not large enough to do between-group analyses (Figure 7). We differentiated botulinum toxin A injections and surgery, but not the specific muscles or type of surgery, as this would have yielded too many possibilities to allow group allocation. This information can be found in Appendix A. One child did not have any intervention on either side, one child did not have any intervention on the right side, and two children had no intervention on the left side. Ten children had only botulinum toxin A injections on both the right and left sides, four children had botulinum toxin A injections and lower limb surgery on both the right and left sides. Seven children had surgery only on the right, six on the left side. For illustration purposes, we merged the right and left legs in Figure 7.

## 4. Discussion

Despite the growing body of literature exploring kinematic patterns and clinical assessments in children with CP, there remains uncertainty about how these factors are inherently associated and whether those associations change longitudinally. Enhanced understanding of these associations could lead to preventive interventions to counteract the development of decreased passive range of motion and increased flexion during gait, as this is known to decrease over time in children with CP [20,23]. Therefore, the aim of this study was to explore the influence of musculoskeletal constraints on gait patterns. More specifically, we explored associations between passive knee extension and hip, knee, and ankle sagittal plane gait kinematics in children with spastic CP assessed on two occasions. Additionally, we compared sagittal plane kinematics longitudinally between two timepoints to explore how gait patterns change over time. By combining associations between physical and kinematic measures with longitudinal assessments, our results provide unique and new information on the relation between musculoskeletal constraints and gait kinematics throughout childhood.

### 4.1. Associations

We found significant but moderate associations between maximal passive knee extension and sagittal knee kinematics in the majority of the stance phase for our full cohort of children with CP and the bilateral CP group. Moreover, if there was a significant association between the joint angles and the passive knee extension at timepoint T1, this association became stronger at the second timepoint for all results. For the total group, this association remained moderate, whereas for the bCP group, r obtained a value of −0.76, quantified as a high correlation. The association between maximal passive knee extension and the knee flexion angle throughout stance in our cohort corroborates the previous findings of Papageorgiou and colleagues, who additionally reported a significant association in terminal swing [22]. Notably, our cohort showed this terminal-swing association at the second timepoint, but not at the first. More interestingly, and reported for the first time, is the increase in the value of the correlation coefficient between timepoints, both for the total group and the bCP group. R obtained its strongest value (~−0.58) between 56 and 60% of the gait cycle at timepoint T1. At timepoint T2, however, R not only increased in magnitude (from ~−0.58 to ~−0.72) but was also reduced (R < −0.60) for a much longer period throughout the stance phase (~20–60%). This longer and stronger association may indicate that only extracting the maximal knee extension value from the whole stance phase might confound important information from earlier stance phase kinematics. From a clinical perspective, if we would like to know how much maximal passive knee extension could possibly contribute to sagittal plane knee kinematics, r^2^ can be considered in addition to r. For the bCP group at timepoint T2, r^2^ ranges between 0.34 and 0.58. This implies that thirty to fifty percent of the sagittal plane knee kinematics could be explained by maximal passive knee extension. It is thus important to emphasize that there are other factors, such as, e.g., balance and other sensory disturbances, that additionally contribute to reduced knee extension throughout stance.

When looking at the discrete values, the mean passive lack of knee extension during the clinical examination was −0.40 degrees for the total group and −0.56 degrees for the bCP group. In stance, the total and bCP group showed a 9- and 10-degree lack of extension in stance, respectively. The increase in the association between maximal passive knee extension and the knee flexion/extension angle throughout stance might imply that a flexed knee gait pattern could possibly be one of the contributing factors to knee contracture development. A flexed knee gait pattern that exceeds the available passive knee extension has been described earlier and requires exploration of the cause of the flexion, which is in agreement with Bartonek et al. [31]. Our currently reported associations are, however, non-directional, so a regression analysis using a larger sample size could possibly confirm this hypothesis. Adding to this theory is the information in Figure 4, indicating that from all individuals who do not yet have a knee contracture at timepoint T1 but will develop one at timepoint T2, none obtain a full knee extension throughout the stance phase at T1. Most children with a knee contracture at timepoints T1 and T2 had bilateral CP and were functioning at GMFCS II. From the five individuals who did not have a knee contracture at T1 but developed one at T2, one was classified as GMFCS I, three were II, and one was III. Thus, the contribution of functional constraints to a flexed knee gait pattern requires further exploration and could benefit from an analysis with more individuals in each functional level.

The lack of association between maximal passive knee extension and ankle plantar/dorsiflexion is in agreement with the findings of Papageorgiou et al. in a larger cohort [18] and could potentially be attributed to the high variability in ankle position in stance. This variability of ankle position, ranging from plantarflexion to dorsiflexion during gait, was visualized and described earlier in this population [32]. In our study, the most notable change between timepoints T1 and T2 was a slight increase in ankle dorsiflexion in mind-stance for individuals with a knee contracture, emphasizing a possible relationship between knee flexion and ankle dorsiflexion during walking in some individuals with CP. These findings corroborate the clinical phenomenon of children ‘sinking’ into a crouch gait pattern by age. Hence, a flexed knee gait in CP has been described as a complex multidimensional deformity with varying natural history [33]. A combination of causes, such as growth spurts, weight gain, and weak quadriceps, is often considered, and sensory aspects additionally require attention, especially in bilateral CP [31,34].

We found significant but moderate associations between maximal passive knee extension and sagittal hip kinematics in the majority of the stance phase for our full cohort of children with CP and the bilateral CP group at T2, but not at T1. The timing of these associations for the total group corroborates previously published results, indicating that short or overactive hamstring muscles spanning both the hip and knee joint could be a contributor to both reduced passive knee extension and reduced hip extension throughout stance [22]. However, the influence of hamstrings lengthening on knee position has been hard to prove [35]. Notably, our bCP group only showed a significant correlation between hip angle and maximal passive knee extension at T2, with a tendency towards significance at T1 (*p* = 0.045). The greater variability in the children with normal passive knee extension (pKE+; Figure 5) may have contributed to the appearance of this significant association.

The short-term influence of pre-operative knee contractures on gait kinematics and functional outcomes was previously assessed by comparing two intervention-matched groups with and without knee contraction before surgery [36]. Both groups improved significantly on the gait deviation index—a measure of gait pathology based on 15 kinematic features at the level of the pelvis, hip, knee, and ankle [37] –, knee kinematics, and step length, indicating that reduced knee extension during gait should be considered when planning interventions, since all included children walked with reduced knee extension in stance. Our findings showed a similar pattern, with visibly greater knee flexion throughout stance at timepoint T2, regardless of any intervention.

The importance of the knee—both passively and during gait—as a predictor for long-term mobility and functional outcomes has long since gained awareness. Graham and colleagues recently validated four developmental stages of musculoskeletal pathology in children with CP: I) hypertonia and dynamic contractures, II) fixed contractures, III) contractures combined with torsional deformities, and IV) decompensation of the musculoskeletal system with more severe musculoskeletal pathology [38]. This classification was used by Thomason et al. to propose a Knee Surveillance program in CP, including clinical history, physical examination, and observational gait analysis to ensure early and appropriate intervention [39]. The main facet of this clinical evaluation is the Simple Knee Contracture Test, which agrees with our maximal passive knee extension assessment. The increase in the magnitude of association values over time corroborates the Knee Surveillance method and emphasizes the importance of regular assessment points to intervene early.

### 4.2. Kinematic Patterns over Time

When comparing sagittal plane kinematic gait patterns over time, our cohort showed no change in knee angle, increased hip flexion throughout stance, and reduced ankle plantarflexion in swing at T2 compared to T1. Only Daly and colleagues have previously reported the longitudinal changes in sagittal plane joint angles using SPM. For the hip angle, the significant differences we reported between timepoints T1 and T2 were in agreement with the findings in Daly’s cohort [20]. In contrast to Daly’s findings, we did not find a significant difference between timepoints for the knee angle. The combination of a slightly greater mean cohort age at both timepoints and a shorter time interval between assessments in the current study may have played a role in the lack of significant differences at the knee level. Moreover, most children who developed a knee contracture at timepoint T2 already had reduced knee extension during gait at timepoint T1. This could indicate that we did not find any significant over-time differences in the sagittal knee flexion/extension angle because various intervention approaches may aim to minimize further knee flexion. At the level of the ankle, the main findings were reduced ankle dorsiflexion throughout stance and reduced plantarflexion throughout swing at timepoint T2 compared to T1, with only the latter obtaining significance. In Daly’s cohort, participants showed a significant difference in stance, but not in swing. The combination of a slightly less impaired cohort in Daly’s study and an older cohort in our study may have contributed to these differences. Studies focusing on repeated gait assessments with fixed time intervals could possibly give us better insights into the change in kinematics at each joint level, especially when stratified according to GMFCS level and the presence of knee contractures [23]. Another possible hypothesis for the paradoxical finding of stable knee kinematics at more limited knee extension with increased hip flexion and decreased dorsiflexion could be reflected as compensatory mechanisms for retaining a steady level of knee extension during the stance phase. However, the role of other factors, including biomechanical factors, such as passive range of motion in the adjuvant joints or growth-related changes, could play significant roles in the complexity of movement patterns during gait in CP [40]. Naturally, many children with repeated gait assessments will have some sort of intervention between assessments, aiming to improve their walking ability and functional constraints. Accordingly, intervention-specific information can be stratified by subgroups to assess the subsequent intervention effects on gait. This is not as straightforward as it sounds, since many children receive multiple interventions of various kinds, and stratifying by every single intervention would limit our outcomes to single-case studies. Therefore, more international collaboration and data merging, including the importance of kinematic patterns on functional aspects as walking speed, step length, and walking distance, would provide opportunities to assess long(er)-term outcomes in larger cohorts.

The effect of intervention methods on the sagittal knee angle during gait is variable (Figure 7), but overall, individuals who received botulinum toxin A in the lower leg muscles showed a notable increase in knee flexion between assessments. Papageorgiou and colleagues explored gait-pattern-specific responses after botulinum toxin A and found only significant improvement of knee extension for the apparent equinus group, but not for the individuals classified according to other gait patterns [14]. However, most children included in the study were high-functioning individuals, classified at GMFCS I, in comparison with our cohort, which also included children at GMFCS levels II and III. Peeters et al. additionally reported no significant short-term (8 weeks after intervention) changes in knee kinematics after botulinum toxin A injections in the gastrocnemius muscles [41]. The effect of botulinum toxin A is, in general, variable and may not be visible in sagittal gait kinematics. However, in combination with either serial casting or physiotherapy interventions, botulinum toxin A may be an effective method to maintain and improve walking ability and delay surgery [42].

Whilst providing important insights into the relation between maximal passive knee extension and sagittal plane gait kinematics over time, the current study has some limitations. Firstly, the mean cohort age at T2 was 12.4 years, which is fairly low. More longitudinal differences could have possibly been obtained if all children at T2 were over 12 years of age. Secondly, while extensive efforts are made by the hospital staff to ensure standardized settings during clinical gait analyses, the timespan between the earliest and most recent gait analysis performed may have caused variability that we cannot control for. Thirdly, our cohort was a mix of patients who came for their first gait analysis (usually between the ages of four and six) and patients who were re-invited before and after an intervention. Whilst this heterogeneity reflects the CP population, it also increases the variability in gait patterns, thereby likely reducing between-group or longitudinal effects. Moreover, other factors such as balance, energy consumption, walking speed, and pain can influence gait patterns and quality and should be considered in future studies to fully understand the complexity of movement and posture in CP.

## 5. Conclusions

Our results showed that the associations between maximal passive knee extension and gait are complex and require investigation beyond biomechanical constraints to elucidate the interactions between these two factors, especially considering that their combination is an important indicator for early and accurate treatment management during interdisciplinary patient care.

## Figures and Tables

**Figure 1 jcm-14-08567-f001:**
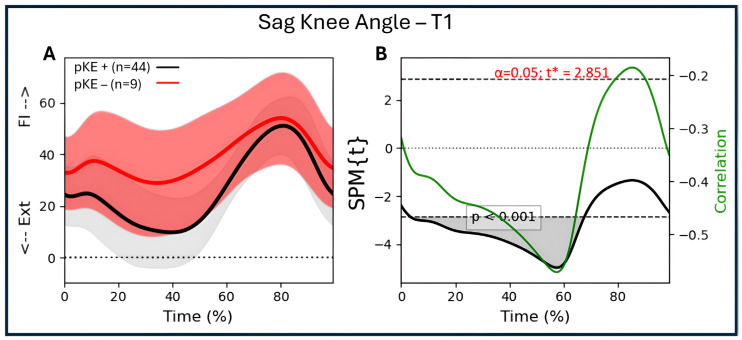
Illustration of the association between maximal passive knee extension and mean angular waveforms of knee flexion/extension during a gait cycle using regression analysis. (**A**) Mean knee flexion/extension angle during gait (time-normalized to 100%) for all individuals with CP. The red group represents the individuals with reduced maximal passive knee extension (pKE −), with the red line representing the mean value and the pink bands the mean plus/minus one standard deviation. The gray group represents individuals with normal maximal passive knee extension (pKE +), with the black line representing the mean value and the gray bands the mean plus/minus one standard deviation. The division between individuals with and without reduced maximal passive knee extension is for illustrative purposes. In the analysis, we included the whole CP as one group and used the value of maximal passive knee extension as the discrete variable to which the sagittal plane gait kinematics were correlated. (**B**) The black line exceeds the critical threshold (dotted lines), meaning that there is a significant association between 3 and 68% of the gait cycle (grey area; one suprathreshold cluster; *p* < 0.001). The green line (y-label on the right side of the figure) indicates the strength of the association (correlation coefficient; R) and its variation throughout the gait cycle. During the period of significant association (3–68%), the correlation coefficient varies between −0.47 and −0.56. Ext = extension; Fl = flexion.

**Figure 2 jcm-14-08567-f002:**
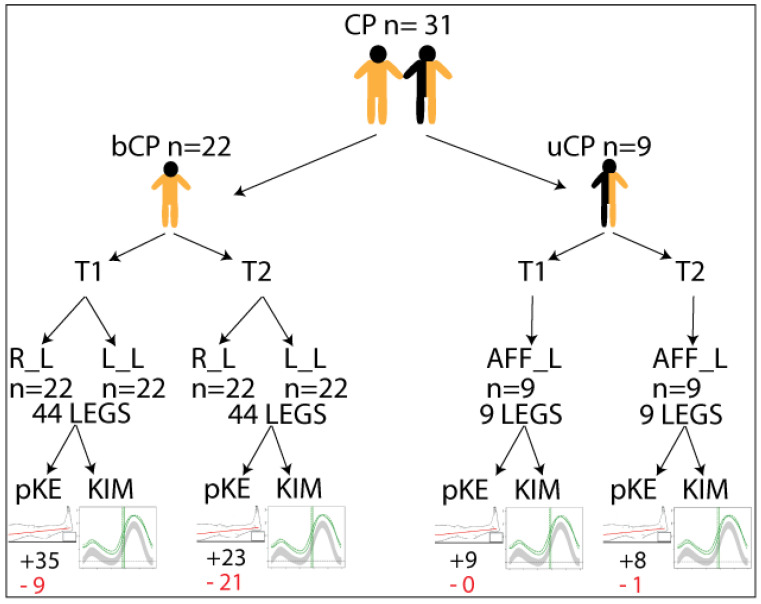
Schematic representation of the included individuals with cerebral palsy (CP) with respect to bilateral (bCP) or unilateral (uCP) and timepointss (T1 and T2) and legs included in the analysis: Right (R_L), Left (L_L), or affected (AFF_L). +/− indicates the number of legs with maximum passive knee extension (pKE) above (black letters)/below (red letters) a neutral position of the knee joint at physical examination, and kinematics (KIM) derived from three-dimensional gait analysis.

**Figure 3 jcm-14-08567-f003:**
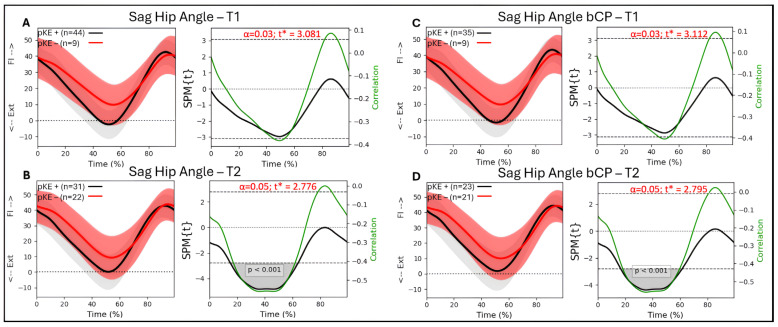
Sagittal hip angle trajectories during a gait cycle derived from three-dimensional gait analysis for individuals with cerebral palsy with reduced maximal passive knee extension (pKE−) in red and normal maximal passive knee extension (pKE+) in black and the associations with maximum passive knee extension (pKE) above (+) or below (−) a neutral position of the knee joint at clinical examination. The whole group of children with spastic CP (n = 41) at (**A**) timepoint T1 and (**B**) timepoint T2, and the bCP group (n = 22) at (**C**) timepoint T1 and (**D**) timepoint T2.

**Figure 4 jcm-14-08567-f004:**
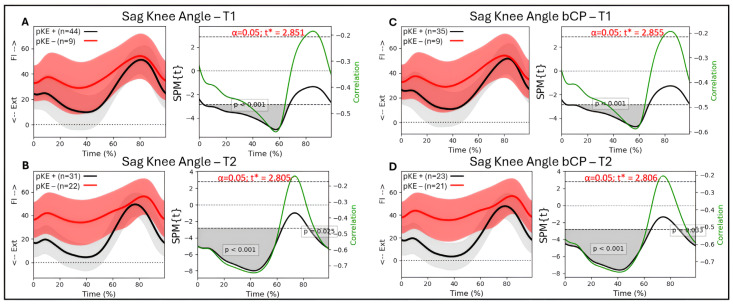
Sagittal knee angle trajectories during a gait cycle derived from three-dimensional gait analysis for individuals with cerebral palsy with reduced maximal passive knee extension (pKE−) in red and normal maximal passive knee extension (pKE+) in black and the associations with maximum passive knee extension (pKE) above (+) or below (−) a neutral position of the knee joint at clinical examination. The whole group of children with spastic CP (n = 41) at (**A**) timepoint T1 and (**B**) timepoint T2, and the bCP group (n = 22) at (**C**) timepoint T1 and (**D**) timepoint T2. Ext = extension; Fl = flexion.

**Figure 5 jcm-14-08567-f005:**
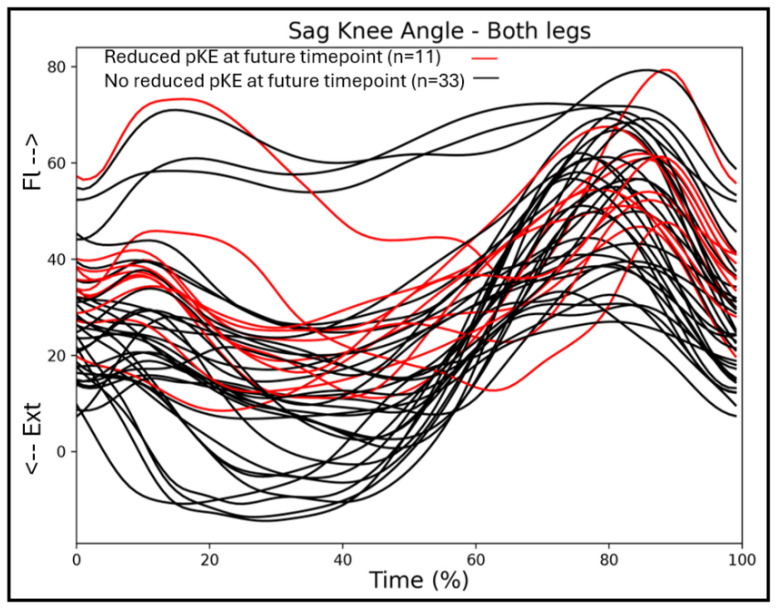
Sagittal knee angles of the children with bilateral CP at timepoint T1, with the trajectories of individuals who will develop a knee contracture at timepoint T2 in red.

**Figure 6 jcm-14-08567-f006:**
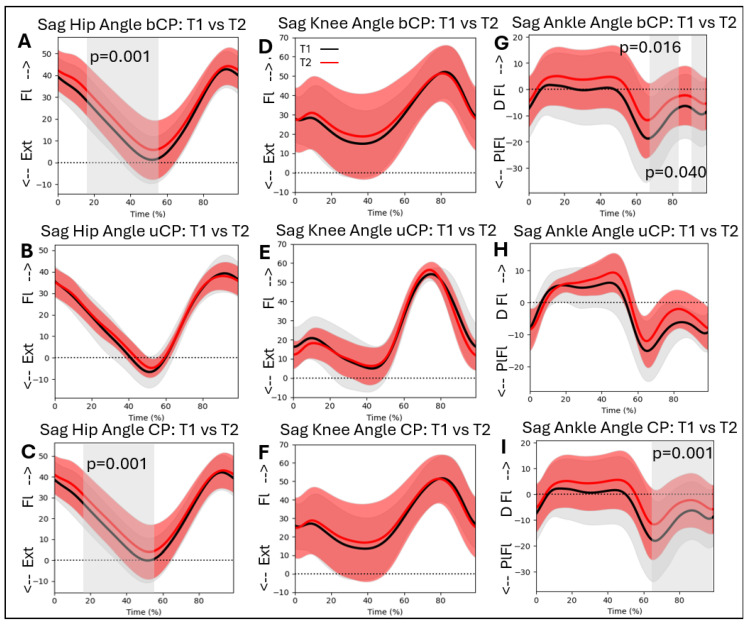
Sagittal hip, knee, and ankle kinematics for the bCP (**A**,**D**,**G**), uCP (**B**,**E**,**H**), and total (**C**,**F**,**I**) groups at timepoints T1 (in black) and T2 (in red). Ext = extension; Fl = Flexion; PlFl = Plantarflexion; DFl = Dorsiflexion.

**Figure 7 jcm-14-08567-f007:**
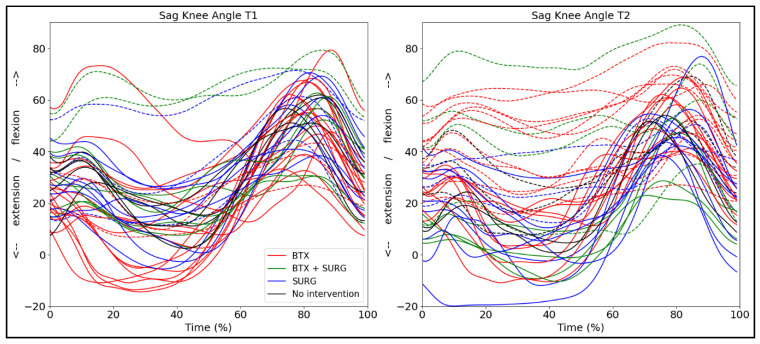
Sagittal knee angle trajectories for both legs (n = 44) of the bCP group differentiated by interventions between T1 and T2: BTX = only botulinum toxin A injections; BTX + SURG is botulinum toxin A injections and any type of surgery; SURG = only surgery. The dotted lines represent the legs that show a passive knee range of motion deficit.

**Table 1 jcm-14-08567-t001:** Participants’ characteristics.

	Sex	Age T1	Age T2	GMFCS	Max pKE		Max Knee Ext in gait	
					T1	T2	T1	T2
TOTAL	M = 29; F = 12	8.2 (2)	12.4 (2.8)	I =14; II = 14; III = 3	R: −0.4 (5.8) L: −0.6 (4)	R: −5.4 (10.7) L: −3.7 (10.4)	R: 12.1 (16.5) L: 8.12 (14.4)	R: 16.4 (22.7) L: 11.5 (16.8)
BCP	M = 13; F= 8	7.7 (1.9)	11.9 (2.8)	I = 6; II = 13; III = 3	R: −0.5 (6.3) L: −0.7 (4.4)	R: −5.9 (11.6) L: −5.2 (11.4)	R: 12.9 (12.2) L: 10.3 (8.1)	R: 19 (22.4) L: 13.2 (20.7)
UCP	M = 5; F = 4	9.3 (1.7)	13.6 (2.4)	I = 8; II = 1	0 (0)	0.5 (4.6)	2.8 (9.9)	3.16 (6.7)

Age, maximal passive knee extension (Max pKE; – = knee extension deficit), and maximal knee extension in gait (Max knee ext) are reported in mean and (standard deviation). BCP = bilateral CP; UCP = unilateral CP; R = right; L = left; GMFCS = Gross Motor Function Classification System; ROM = range of motion; Max knee Ext = maximal knee extension in stance phase. T1 = assessment at timepoint 1; T2 = assessment at timepoint 2. For the UCP group, only the values for the affected leg are reported.

## Data Availability

The datasets presented in this article are not readily available due to the sensitive nature of the included data, in order to protect the participants’ anonymity.

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
