# Peer review of "Associations Between Maximal Passive Knee Extension and Sagittal Plane Kinematic Patterns in Children with Spastic Cerebral Palsy: A Longitudinal Study"

_jcm, 2025, doi:10.3390/jcm14238567_

Round 1

Reviewer 1 Report

Comments and Suggestions for Authors

This is an interesting longitudinal study exploring the relationship between passive knee extension range of motion and gait kinematics in children with spastic cerebral palsy. It has some limitations that reduce the quality of the methodology, such as the heterogeneity of the participants (bilateral vs. unilateral, different GMFCS) and the absence of a healthy control group to compare physiological changes specific to development.

Introduction

  • No detailed analysis of previous studies on the pKE-walking association is provided.
  • References to the influence of common interventions (orthopedic surgery, intensive physical therapy, botulinum toxin) are lacking, which are highly relevant for interpreting longitudinal changes.

Methods

  • Covering the period from 2008 to 2025, it is likely that assessment protocols, gait analysis equipment, and treatment strategies have changed, which may introduce uncontrolled variability. This should be justified and discussed.
  • Although the pKE measurement was performed in a standardized manner, intra-/interobserver reliability is not detailed; this is relevant because it is a clinical measurement that depends on the examiner.
  • Although controlled for age, sex, and GMFCS, other important variables (use of orthotics, intensive physical therapy, botulinum toxin, minor surgeries) were not included.

Discussion

  • The discussion could delve deeper into the clinical relevance of the changes observed (e.g., what difference in degrees of extension translates into a noticeable improvement in gait?).

Author Response

Reviewer 1

This is an interesting longitudinal study exploring the relationship between passive knee extension range of motion and gait kinematics in children with spastic cerebral palsy. It has some limitations that reduce the quality of the methodology, such as the heterogeneity of the participants (bilateral vs. unilateral, different GMFCS) and the absence of a healthy control group to compare physiological changes specific to development.

Introduction

  1. No detailed analysis of previous studies on the pKE-walking association is provided.

Reply: We thank the reviewer for this remark. We have added more specific information on the correlation values from the two studies available in the introduction section (Desloovere et al.; McMulkin et al.). The current sentence now reads [p.2 line 79-83]:

While the importance of 3DGA for clinical decision making and assessment of interventions has been well established, previously reported associations between clinical assessments and parameters extracted from 3DGA were low [15,16]. Desloovere et al. found that correlations between passive hip and knee range of motion and maximal knee and ankle angle during different phases of the stance phase ranged from -0.14 to 0.50 [15]. Similarly, McMulkin and colleagues reported associations between passive knee extension and knee extension at initial contact and minimal knee flexion of -0.24 and -0.45, respectively [16].

  1. References to the influence of common interventions (orthopedic surgery, intensive physical therapy, botulinum toxin) are lacking, which are highly relevant for interpreting longitudinal changes.

Reply: Thank you for highlighting this. We have added more information and references on common intervention outcomes on longitudinal changes in kinematic patterns. The new paragraph in the introduction now reads [p.2 line 66-67]:

Various interventions have shown positive effects on passive joint range of motion, kinematic patterns. Ounpuu and colleagues assessed the long-term (11 year) effects of multilevel surgery in 51 adults with CP and found that knee flexion at initial contact and peak ankle dorsiflexion in stance were maintained, but range of motion improvements were not maintained beyond one-year post-surgery [11]. Verreydt et al. found no range of motion improvements one year after a selective dorsal rhizotomy in 15 children with CP, but the total gait profile score and the gait profile score at the level of the ankle and knee did improve post-intervention. The reported effects of botulinum-toxin A on clinical and gait parameters are variable, ranging from no effects [12] to significant improvements in knee position in midstance [13] and short-term improvements in kinematics patterns of the ankle and the knee [14].

Methods

  1. Covering the period from 2008 to 2025, it is likely that assessment protocols, gait analysis equipment, and treatment strategies have changed, which may introduce uncontrolled variability. This should be justified and discussed.

Reply: Thank you for this comment that is important for interpretation of the results. At the Motion analysis laboratory at Astrid Lindgren Children’s hospital, we are using the 3D motion analysis provided by Vicon. Assessment protocols, marker set up, recordings and processing of data are following the standard of European Society of Motion Analysis (ESMAC). Equipment is updated consciously, but the marker set was not modified in the period of assessments in the current study. New Vicon cameras were, however, obtained in 2019, yielding 12 cameras instead of eight, providing even more precise data. The lab is for research use and for clinical use with most of the staff involved in both and there is a close collaboration between research and clinical work. All staff attend international, European and Nordic conferences on a regular basis to ensure high competence. In the group, we have regular joint sessions regarding marker placement, where inter- and intra rater reliability are assessed on a visual basis. To clarify this, the following text has been added in the ‘Three-dimension gait data’ section [p.5 line 193-197]:

To strive for valid and reliable data and limit the risk for uncontrolled variability, routines for gait analysis recordings, processing of data and clinical examinations are standardized following the standard of European Society of Motion Analysis. Equipment has been updated with new and more cameras during the period of data collection.

  1. Although the pKE measurement was performed in a standardized manner, intra-
    /interobserver reliability is not detailed; this is relevant because it is a clinical
    measurement that depends on the examiner.

Reply: Thank you for the comment. The physiotherapists performing the clinical examinations including measurements of pKE have special training following a standardised protocol before being independent in the lab. In the group, we have joint sessions regarding the protocol for clinical examinations with inter- and intra-rater reliability testing of on a regular basis. Most physiotherapists working in the lab have a PhD, are doctoral students or have a degree of Master of Science. To clarify this, the following has been added in the Materials and Methods section [p.4 line 169]:

All participants underwent a clinical examination including measurements of maximum passive range of motion of hip, knee and ankle with a goniometer in a supine position by specially trained, experienced physiotherapist following a standardized protocol [25].

  1. Although controlled for age, sex, and GMFCS, other important variables (use of orthotics, intensive physical therapy, botulinum toxin, minor surgeries) were not included.

Reply: Thank you for the remark that is covering important research questions. In future studies including more participants, aspects of interventions such as use of orthotics, intensive physical therapy, botulinum neurotoxin A, minor surgeries and their effect on kinematics will hopefully be explored. That research question will however require another research protocol, and we will therefore not focus on specific intervention methods in this manuscript. For completeness, I have added the interventions per intervention group from the participants in the current study in Supplementary Table 1 [p.18]. Due to privacy regulations, we are not allowed to share more detailed intervention information.

Discussion

  1. The discussion could delve deeper into the clinical relevance of the changes observed (e.g., what difference in degrees of extension translates into a noticeable improvement in gait?).

Reply: Thank you for this comment. Considering the feedback of all reviewers indicating caution as to not interfere associations with cause and effect, I will refrain from drawing clinical conclusions based on the results of the current study, as these claims may be overstating our findings.

Reviewer 2 Report

Comments and Suggestions for Authors

The introduction appears to provide sufficient background to support the need for this research; the purpose statement needs to be rewritten to be more specific. Methods was organized well but more technical information is needed; I have suggested a way to enhance the quality of your correlation analysis. Results and discussion are well organized and consistent with the study's purpose. Some revision is needed for tables and figures in the results section. The conclusion was appropriately conservative given the nature of the study design. There were several grammatical, word choice, and word tense errors that need to be corrected. An English grammar consultant may be warranted. Please check all references cited for adherence to journal instruction and consistency. See specific comments in the pdf.

Comments on the Quality of English Language

See comments above.

Author Response

Reviewer 2

The introduction appears to provide sufficient background to support the need for this research; the purpose statement needs to be rewritten to be more specific. Methods was organized well but more technical information is needed; I have suggested a way to enhance the quality of your correlation analysis. Results and discussion are well organized and consistent with the study's purpose. Some revision is needed for tables and figures in the results section. The conclusion was appropriately conservative given the nature of the study design. There were several grammatical, word choice, and word tense errors that need to be corrected. An English grammar consultant may be warranted. Please check all references cited for adherence to journal instruction and consistency. See specific comments in the pdf.

Reply: As the reviewer provided good comments on the content and grammar of the manuscript in a PDF document, I have copied the comments that provide feedback on the content of the manuscript here to assure a comprehensive overview of all changes for the reviewers. I have not copied all requests for grammatical changes, but I have modified these directly in the revised version of the manuscript.

Introduction

  1. I'm not sure that SPM is suited to analyze complex relationships; please consult a statistical expert on this

Reply: SPM is currently the only method available to analyse time-dependent input data. Since we were specifically interested in the relationship between the passive knee extension and the kinematic pattern throughout the gait cycle, this method is most suitable. However, we acknowledge the need to emphasize that we are only exploring associations and we cannot make any assumptions about inference, as kindly noted by reviewer 3. We have adjusted our wording throughout the manuscript based on careful consideration considering the applied methodology. Please refer to the comment of reviewer 3 for the specific text modifications.

  1. “The primary objective is to explore the associations between maximal passive knee extension and sagittal plane joint kinematics throughout the gait cycle at two timepoints during childhood for each individual with CP.” yet you also report hip kinematics; the purpose statement should be revised to more accurately state what was done and why

Reply: Indeed, the purpose was to assess the associations between passive knee range of motion and the kinematic patterns of the hip, knee and ankle. This is because we know that reduced maximal passive knee extension will not only influence the knee motion during gait, but its effects may also extend to the ankle and the hip. We have adjusted the introduction to better reflect this thought. The adjusted section of the introduction now reads [p.3 line 120-127]:

In children with bilateral CP (bCP), lack of passive knee extension was associated with decreased knee extension and decreased ankle dorsiflexion throughout stance, whereas this was not the case in children with unilateral CP (uCP) [22]. The association at the level of the ankle indicates that reduced passive knee extension could be a contributing factor to deviating patterns that go beyond the level of the knee.   

We have also re-written the objective to clarify this [p.4 line 139-147]:

Considering the importance of growth and development and the lack of information about longitudinal relationships between passive joint range of motion and kinematic angles at different joint levels during walking, current study aimed to associate clinical assessments and gait kinematics using longitudinal assessments. The primary objective was to explore the associations between maximal passive knee extension and sagittal plane joint kinematics of the hip, knee and ankle throughout the gait cycle at two timepoints during childhood for each individual with CP.

  1. “Sagittal plane gait data of children with spastic CP who visited the Motion analysis Laboratory at Karolinska University Hospital for a clinical gait analysis between 2008 and 2025 was included.

 this sounds like this study is a retrospective design...if so, please make this explicit

Reply: While we retrospectively used the data that was collected for clinical purposes, some of the data was collected after obtaining ethical approval. Therefore, the MDPI editor requested us to remove the word ‘retrospective’ from the methods section, to which we adhered.  

Methods

  1. Please provide details on camera capture rate

Reply: We have re-written this whole section to make the workflow and technical details clearer. The section now reads [p.4/5 line 173-187]:

During the three-dimensional gait analysis, the child walked barefoot on a 10m walkway at a self-selected walking speed until three trials with good force plate hits were obtained for clinical purposes. However, force plate information was not used in the current study. Reflective markers (diameter: 10mm) were attached to the skin according to the Plug-In Gait lower limb model using 16 reflective markers placed on anatomical landmarks [26-28]. Marker trajectories were sampled at 100Hz with a 8 to 12 Vicon camera system (Vicon-UK, Oxford, United Kingdom). A built-in Woltring filter using mean square error and a 10 mm² smoothing factor was used to filter the trajectories. Gait cycles were identified from initial contact and foot-off events, and hip, knee and ankle kinematics were calculated in the Vicon Nexus software.

  1. Please report your intra- and inter-rater reliability for gait kinematics since your design is a test-retest design and since you imply above that more that one therapist was involved; if the therapists did not place markers you need to specify who placed markers and if more than one person participated in this; if you do not have your own test-retest reliability data you need to provide references regarding reliability using the lower limb PIG model

Reply: Thank you for this comment. As mentioned in the reply to to comment of reviewer 1, all physiotherapists performing the clinical examinations including measurements of pKE have special training following a standardised protocol before being independent in the lab. We additionally have joint sessions regarding the protocol for clinical examinations with inter- and intra-rater reliability testing of on a regular basis. I have added a reference on intra- and inter-rater reliability of the PiG model as we do not have a reference for our specific lab, please see references in the reply of the comment above. Even if we had inter- or intra-rater reliability available for our lab, it would be impossible to tell whether the current differences are due to rater inconsistencies or actual treatment effects, since the majority of our patients had treatment between the two assessments.

  1. How many walking trials were collected for each participant? Were force plate used? If force plates were used, specify data collection frequency.

Reply: Thank you noting that this was not clearly described. The number of trials commonly collected in clinical is three walking trials with good force plate hits. The number of trials each individual performed is thus not consistent, as it depends how many trials they need to obtain successful force plate hits. However, force plate information was not included in the current study. We have added this information in the methods section, which now reads [p.4 line 174-176]:

During the three-dimensional gait analysis, the child walked barefoot on a 10m walkway at a self-selected walking speed until three trials with good force plate hits were obtained for clinical purposes. However, force plate information was not used in the current study.

  1. Please specify what was used to define gait cycles

Reply: We apologise for not making this clearer. The gait events (initial contact and foot off) were used to define the gait cycles. We have added this information and the sentence now reads [p.5 line 180-182]:

Gait cycles were identified from initial contact and foot-off events, and hip, knee and ankle kinematics were calculated in the Vicon Nexus software.

  1. You need to provide more information and data reduction (how data was filtered etc.). It would also be helpful to provide a figure showing the lab setup that includes all instrumentation, i.e, cameras, force plates, video cameras and includes the orientation of the calibrated laboratory coordinate system.

Reply: Thank you for this remark. It is uncommon to include a figure of the lab set-up – none of the studies that we mention in our reference list have included a figure of their lab set-up. However, if the reviewer insists, we can include a picture of our lab in the Supplementary Material. Regarding filtering: We have added the filtering information in the methods section. The sentence now reads [p.4/5 line 179-188]:

Marker trajectories were sampled at 100Hz with a 8 to 12 Vicon camera system (Vicon-UK, Oxford, United Kingdom). A built-in Woltring filter using mean square error and a 10 mm² smoothing factor was used to filter the trajectories.

Results

  1. A passive joint angle really means that the knee is flexed, correct? somewhere you need to state clearly your expressed conventions.

Reply: Thank you for noting that this needs clarification. We have added a sentence to clarify this convention. The sentence now reads [p.8 line 277-278]:

In accordance with gait analysis standards, full knee extension is zero degrees, positive values reflect knee flexion and negative values reflect knee hyperextension.

  1. Gait vs kinematics patterns

Reply: In accordance with the reviewer’s suggestion, the titles ‘Gait patterns’ have been modified to ‘Kinematic patterns’ everywhere in the manuscript.

  1. [In the section ‘Kinematic patterns with respect to interventions’] In which muscles has botulinum neurotoxin A been injected and what type surgeries have been performed?

Reply: Thank you for this important remark, which is similar to the remark of reviewer 1. Therefore, we refer to p.1 of this rebuttal for our reply to this remark.

Discussion

  1. Prior to restating the purpose also 1) briefly describe the problem, i.e., gap in the literature, you attempted to address and 2) the rationale/need and clinical importance of this research

Reply: Thank you for this advice. We have re-written the start of the discussion section to include these elements. The section now reads [p.12 line 363-369]:

Despite the growing body of literature exploring kinematic patterns and clinical assessments in children with CP, there remains unclarity about how these factors are inherently associated and whether those associations change longitudinally. Enhanced understanding of these associations could lead to preventive interventions to counteract development of decreased passive range of motion and increased flexion during gait, as this is known to  decreases over time in children CP [23] [20]. Therefore, the aim of this study was to explore the influence of musculoskeletal constraints on gait patterns.

Conclusion

  1. I need to make comments on the use of a design that is primarily based on correlation analysis. First, although I did not see evidence of this, one must avoid making any inferences about cause effect. You rightly included a scale of different magnitude r values which is useful to use to complement statistical significance; using and discussing the relative strength of different r values, i.e., poor, fair, etc., provides, perhaps, the clinical relevance of the findings. Unfortunately, you did not do this and you should have. For example, even your largest value, e.g., ~0.7, can only explain about 50% of the variance when you look at it using the coefficient of determination (r^2); since so much of your results rely on r, I strongly recommend you use the scale described in the stats section of methods, and complement this by calculating r^2 as well. Although you may not present this in the results section you MUST discuss this since it can add to the clinical relevance of your findings. You might consult a statistical expert to verify my contention.

Reply: We thank the reviewer for this rightful suggestion. We have modified the discussion section to include interpretation of r according to the scale mentioned in our methods section, as well as r^2. The revised sections now read [p.12/13 line 378-402]:

4.1 Associations

We found significant but moderate associations between maximal passive knee extension and sagittal knee kinematics in the majority of stance phase for our full cohort of children with CP and the bilateral CP group. Moreover, if there was a significant association between the joint angles and the passive knee extension at timepoint T1, this association became stronger at the second timepoint for all results. For the total group, this association remained moderate whereas for the bCP group, r obtained a value of -0.76, quantifying as high correlation. [...] This longer and stronger association may indicate that only extracting the maximal knee extension value from the whole stance phase might confound important information from earlier stance phase kinematics. From a clinical perspective, if we would like to know how much maximal passive knee extension could possibly contribute to sagittal plane knee kinematics, r2 can be consider in addition to r. For the bCP group at timepoint T2, r2 ranges between 0.34 and 0.58. This implies that thirty to fifty percent of the sagittal plane knee kinematics could be explained by maximal passive knee extension. It is thus important to emphasize that there are other factors such as e.g. balance and other sensory disturbances that are additionally contributing to reduced knee extension throughout stance.

References

  1. It looks like some journal titles were abbreviated and others not; please consult journal instructions for references and review all ref for compliance/accuracy and consistency

Reply: Thank you for pointing this out. We used MDPI’s reference style as downloaded from the Journal’s website, but it seems like the formatting did not keep up when changing to the Journal’s template. We have now corrected this.

Reviewer 3 Report

Comments and Suggestions for Authors

The study explores an important and clinically relevant issue: the relationship between passive knee extension and sagittal plane gait kinematics in children with spastic cerebral palsy (CP), and how these relationships evolve over time. The use of longitudinal 3D gait analysis (3DGA) and Statistical Parametric Mapping (SPM) is methodologically appropriate and innovative within this domain. However, while the experimental design is solid and clearly described, the conceptual framing, clinical interpretation, and discussion of implications remain underdeveloped. The manuscript would benefit from tighter focus, a clearer articulation of the study’s novelty, and a more robust discussion of how these findings translate into clinical practice.

Major Concerns

1. Clarity of research question and hypothesis

The introduction (p. 2, l. 20–35) does not clearly state a hypothesis. The text alternates between exploring “associations” and implying causality (“reduced ROM may contribute to flexed gait”), creating ambiguity. The authors should clearly define whether the study tests correlation or causal inference.

2. Underdeveloped clinical significance (“so what”)

The key finding—that knee flexion during gait remains stable despite more contractures—is stated (p. 10, l. 25–32) but not adequately interpreted. The discussion should explain whether this stability reflects adaptive compensation, measurement insensitivity, or other factors. As written, it leaves readers unsure why this matters clinically.

3. Overinterpretation of correlation as causation

Statements suggesting that gait flexion “may drive” contracture development (p. 12, l. 8–14) are speculative and unsupported by the data. The authors should rephrase such sentences as hypotheses rather than conclusions.

4. Sample size and statistical power

The sample (31 children; 53 legs) is small relative to the number of regressions performed. No power analysis is presented (p. 4–5). The authors should discuss statistical power and risk of Type I/II errors, especially given multiple SPM comparisons.

5. Heterogeneity and intervention confounding

The dataset includes children pre- and post-intervention, some with surgery or botulinum toxin (p. 9–10). This variability may mask or exaggerate longitudinal trends. Although acknowledged, it should be more systematically addressed (e.g., subgroup or sensitivity analyses).

6. Limited exploration of compensatory mechanisms

The paradoxical finding (stable gait despite more contractures) likely reflects compensations at the hip and ankle. This plausible mechanism is not analyzed or discussed (p. 11–12). The authors should explore this biomechanical explanation or acknowledge it as a limitation.

7. Missing functional outcomes

No walking speed, energy cost, or participation measures are reported. Without these, it is difficult to relate biomechanical correlations to functional capacity or quality of life (entire study). The omission weakens the clinical translation.

8. Growth and maturation effects

Participants range from 4 to 17 years, yet the manuscript does not control for or discuss growth-related changes (p. 6, l. 15–25). Since joint range and gait patterns vary with age, normalization or discussion of developmental influence is needed.

9. Transparency of AI-assisted analysis

The sentence “GenAI has been used in this paper to assist in data analysis” (p. 5, l. 27–28) lacks clarity. Specify what AI tools were used, for what tasks, and confirm manual verification of all outputs.

10. Clinical translation and surveillance recommendations

The conclusion (p. 14–15) notes that combined clinical and gait data are useful for early management but stops short of defining how this can be operationalized. The paper should propose specific clinical surveillance indicators or follow-up intervals.

Minor Concerns

1. Abstract clarity – p. 1, l. 12–30

The abstract is densely written and overly technical. Simplify sentences and clearly state novelty: “This study demonstrates that even small limitations in passive knee extension correlate with stable but flexed gait, emphasizing the need for early monitoring.”

2. Redundant phrasing in the Introduction – p. 2, l. 15–34

Sentences repeating that the interplay between ROM and gait is “complex and individualized” appear twice. Merge for concision.

3. Inconsistent terminology – p. 4, l. 2–10

Ensure consistent use of abbreviations: pKE, ROM, bCP, and uCP. Occasionally the text alternates between “passive knee extension” and “maximum passive knee extension.”

4. Statistical description – p. 5, l. 10–22

Clarify how the alpha level was corrected for multiple comparisons. The current phrase “adjusted for degrees of freedom of each joint” is vague.

5. Correlation interpretation – p. 7, l. 6–15

Correlations of −0.4 to −0.7 are described as “high.” They should be referred to as moderate according to Hinkle et al. (2003), which the paper itself cites.

6. Discussion organization – p. 11–13

The Discussion is long and descriptive. Work on clarity or add subheadings such as “Associations,” “Clinical implications,” and “Limitations” to guide the reader.

7. Ethics and consent clarification – p. 15, l. 20–25

Specify whether consent for retrospective data use was obtained separately from initial clinical consent.

Author Response

  1. Clarity of research question and hypothesis

The introduction (p. 2, l. 20–35) does not clearly state a hypothesis. The text alternates between exploring “associations” and implying causality (“reduced ROM may contribute to flexed gait”), creating ambiguity. The authors should clearly define whether the study tests correlation or causal inference.

Reply: Thank you for this comment. We have re-phrased this sentence in the introduction to avoid the mentioning of ‘cause and effect’ in order to provide more clarity. The sentence now reads [p.3 line 128-132]:

Papageorgiou et al. included a large sample size with a wide age span, ranging from three to 18 years of age [22]. This broad age range complicates the interpretation of the associations, because the effect of normal growth cannot be modelled in a cross-sectional study design.    

  1. Underdeveloped clinical significance (“so what”)

The key finding—that knee flexion during gait remains stable despite more contractures—is stated (p. 10, l. 25–32) but not adequately interpreted. The discussion should explain whether this stability reflects adaptive compensation, measurement insensitivity, or other factors. As written, it leaves readers unsure why this matters clinically.

Reply: Thank you for noting this. We have added possible hypotheses on why knee flexion in gait remains relatively stable despite the number of knee contractures increasing. The section in the discussion now reads [p.14 line 477-483]:

The combination of a slightly greater mean cohort age at both timepoints and a shorter time interval between assessments in the current study may have played a role in the lack of significant differences at the knee level. Moreover, most children who developed a knee contracture at timepoint T2 already had reduced knee extension during gait at timepoint T1. This could indicate that we did not find any significant over time differences in the sagittal knee flexion/extension angle because various intervention approaches may aim to minimize further knee flexion.

  1. Overinterpretation of correlation as causation

Statements suggesting that gait flexion “may drive” contracture development (p. 12, l. 8–14) are speculative and unsupported by the data. The authors should rephrase such sentences as hypotheses rather than conclusions.

Reply: Thank you making us aware of this overspeculation. We agree that the sentence might cause misinterpretations, and we have accordingly removed the sentence from the discussion section.

  1. Sample size and statistical power

The sample (31 children; 53 legs) is small relative to the number of regressions performed. No power analysis is presented (p. 4–5). The authors should discuss statistical power and risk of Type I/II errors, especially given multiple SPM comparisons.

Reply: Thank you for this comment. Since statistical parametric mapping (SPM) uses random field theory to address the problem of ‘multiple comparisons’, it is not possible to perform a classical power calculation. SPM is based on spatio-temporal correlations between time-points of the continuous outcome variable, meaning that there are fewer independent comparisons than the number of time-points in the signal. Therefore, using SPM as a method inherently reduces the risk of type II errors. However, we agree that the interpretation of the alpha-level could be more stringent considering we did test on the bCP group and the whole group. We have therefore divided the alpha-level by 2 for all tests to further reduce the risk op type II errors. This approach mainly had consequences for the correlations between maximal passive knee extension and hip kinematics at timepoint T1. The methods, results and discussion section have been modified accordingly.

Statistical Analysis Section [p.5 line 209-211]:

The alpha-level for all correlation analyses was set to 0.025 to considering correlations were performed for both the whole group and the bCP group. 

Results Section [p.8 line 286-296]:

3.5. Associations between passive knee extension and hip kinematics 

For the whole group (n=53 legs) and the bCP group (n=44 legs) at T1, there were no significant associations between hip flexion/extension during gait and passive knee extension at T (Figure 3A/C). At T2, there was a significant association between hip flexion/extension and passive knee extension from 15-64% (p<0.001) of the gait cycle, with R between -0.41 and -0.57 for the whole group (Figure 3B). For the bCP group (n=44 legs), there was a significant association between hip flexion/extension and passive knee extension from 17-61% (p<0.001) of the gait cycle, with R between -0.42 and -0.57 at T2 (Figure 3D).

Discussion Section [p.13 line 436-438]:

We found significant but moderate associations between maximal passive knee extension and sagittal hip kinematics in the majority of stance phase for our full cohort of children with CP and the bilateral CP group at T2 but not at T1.

  1. Heterogeneity and intervention confounding

The dataset includes children pre- and post-intervention, some with surgery or botulinum toxin (p. 9–10). This variability may mask or exaggerate longitudinal trends. Although acknowledged, it should be more systematically addressed (e.g., subgroup or sensitivity analyses).

Reply: Thank you for this comment. This agrees with the comment nr.5 of reviewer 1, and I have added a Supplementary Table containing intervention information per intervention group. However, subgroup or sensitivity analysis is impossible due to the varying nature of interventions and multiple combinations per individual. Even when only considering the division ‘surgery’, ‘botulinum neurotoxin A’, ‘surgery + botulinum neurotoxin A’ and ‘no intervention’, we did not have big enough groups to provide a meaningful subgroup analysis. We thus aim to focus on intervention outcomes and their influence on kinematic patterns in a future study, but the current sample size is not big enough to consider this aspect as a main focus of the current study.

  1. Limited exploration of compensatory mechanisms

The paradoxical finding (stable gait despite more contractures) likely reflects compensations at the hip and ankle. This plausible mechanism is not analyzed or discussed (p. 11–12). The authors should explore this biomechanical explanation or acknowledge it as a limitation.

Reply: Part of this comment is similar to comment nr.2 (‘underdeveloped clinical significance’) and we have addressed some of the possible issues in our reply to comment nr.2. We have additionally added a sentence in the discussion regarding possible compensation mechanisms at different joint levels [p.15 line 492-497]:

Another possible hypothesis for the paradoxical finding of stable knee kinematics at more limited knee extension with increased hip flexion and decreased dorsiflexion could be reflected as compensatory mechanisms for retaining a steady level of knee extension during stance phase. However, the role of other factors biomechanical factors such passive range of motion in the adjuvant joints or growth-related changes could play significant roles in the complexity of movements patterns during gait in CP [39]

  1. Missing functional outcomes

No walking speed, energy cost, or participation measures are reported. Without these, it is difficult to relate biomechanical correlations to functional capacity or quality of life (entire study). The omission weakens the clinical translation.

Reply: Thank you for this comment. Including information such as spatio-temporal parameters, energy cost and patient reported information on health in the analysis would enhance understanding on the role of kinematic patterns on walking of importance for the participants. However, functional outcomes were not the focus of current study. We have added this to the limitation section [p.15 line 533-536]:

Moreover, other factors such as balance, energy consumption, walking speed and pain can influence gait patterns and quality and should be considered in future studies to fully understand the complexity of movement and posture in CP.

  1. Growth and maturation effects

Participants range from 4 to 17 years, yet the manuscript does not control for or discuss growth-related changes (p. 6, l. 15–25). Since joint range and gait patterns vary with age, normalization or discussion of developmental influence is needed.

Reply: Thank you for this comment. For the associations, each child’s maximum passive knee extension was associated with hip, knee and ankle kinematics at the same assessment point. More normalization would thus not be appropriate for this method. For the comparison of kinematic patterns between timepoints, growth-related changes could play a role. We have added this in the discussion section [p.15 line 492-397]:

Another possible hypothesis for the paradoxical finding of stable knee kinematics at more limited knee extension with increased hip flexion and decreased dorsiflexion could be reflected as compensatory mechanisms for retaining a steady level of knee extension during stance phase. However, the role of other factors biomechanical factors such passive range of motion in the adjuvant joints or growth-related changes could play significant roles in the complexity of movements patterns during gait in CP [39]

  1. Transparency of AI-assisted analysis

The sentence “GenAI has been used in this paper to assist in data analysis” (p. 5, l. 27–28) lacks clarity. Specify what AI tools were used, for what tasks, and confirm manual verification of all outputs.

Reply: We agree with this comment and we have re-written this sentence to include more details on which GenAI was used and for which tasks. The paragraph now reads [p.5 line 222-225]:

GenAI has been used in this paper to assist in data analysis. More specifically, Microsoft Co-Pilot was used to assist in writing the code to normalize kinematic data and generate the figures included in the manuscript. All outputs from GenAI were manually verified to ensure correct outputs. No GenAI was used in the writing of the manuscript.

Round 2

Reviewer 1 Report

Comments and Suggestions for Authors

Most of the reviewer's suggestions have been addressed.

Author Response

We thank the reviewer for acknowledging that we have sufficiently addressed all comments. If we have understood correctly, there is no request for further improvements.

Reviewer 2 Report

Comments and Suggestions for Authors

The revised manuscript is significantly improved in all sections. I see that you have now included the use of a scale operationally defining the relative strength of correlations. It is also noted that the use of a coefficient of determination can enhance the clinical interpretation of associations. However, you need to describe the use of the coefficient of determination, its definition, and why it is important in the stats analysis section of methods. This is important so that when the reader sees you report r^2 in the results section and discuss it they understand what is going on.

Author Response

We thank the reviewer for the continuation of their feedback and we have addressed the remaining comment:

I see that you have now included the use of a scale operationally defining the relative strength of correlations. It is also noted that the use of a coefficient of determination can enhance the clinical interpretation of associations. However, you need to describe the use of the coefficient of determination, its definition, and why it is important in the stats analysis section of methods. This is important so that when the reader sees you report r^2 in the results section and discuss it they understand what is going on.

Reply: We have added a sentence in the statistical analysis section to address the explanation of r^2 (p. 5 line 215-217).  The sentence now reads:

R2 – the coefficient of determination – was additionally used to interpret the clinical strength of the found associations, as this value indicates the proportion of variance that is shared by two variables [30].